# VISUAL ODOMETRY WITH TRANSFORMERS

## ABSTRACT

Modern monocular visual odometry methods typically combine pre-trained deep learning components with optimization modules, resulting in complex pipelines that rely heavily on camera calibration and hyperparameter tuning, and often struggle in unseen real-world scenarios. Recent large-scale 3D models trained on massive amounts of multi-modal data have partially alleviated these challenges, providing generalizable dense reconstruction and camera pose estimation. Still, they remain limited in handling long videos and providing accurate per-frame estimates, which are required for visual odometry. In this work, we demonstrate that monocular visual odometry can be addressed effectively in an end-to-end manner, thereby eliminating the need for handcrafted components such as bundle adjustment, feature matching, camera calibration, or dense 3D reconstruction. We introduce VoT, short for **V**isual **o**dometry **T**ransformer, which processes sequences of monocular frames by extracting features and modeling global relationships through temporal and spatial attention. Unlike prior methods, VoT directly predicts camera motion without estimating dense geometry and relies solely on camera poses for supervision. The framework is modular and flexible, allowing seamless integration of various pre-trained encoders as feature extractors. Experimental results demonstrate that VoT scales effectively with larger datasets, benefits substantially from stronger pre-trained backbones, generalizes across diverse camera motions and calibration settings, and outperforms traditional methods while running more than $3\times$ faster. The code will be released.

## 1 INTRODUCTION

The goal of monocular visual odometry is to estimate a camera's position and orientation from a sequence of video frames (Cadena et al., 2016). In recent years, monocular visual odometry has gained increasing attention across various fields, including augmented reality, virtual reality, autonomous driving, and robotics. This growing interest is particularly notable when compared to systems that rely on stereo vision (Wang et al., 2017a; Engel et al., 2014) or multimodal inputs, such as visual-inertial odometry (Von Stumberg et al., 2018; Forster et al., 2015). Although monocular setups present greater challenges, their simplicity in deployment enables even broader applicability in real-world scenarios. Therefore, this paper focuses on monocular visual odometry using neural network-based approaches.

There has been rapid progress in monocular visual odometry over a short period, highlighting the potential of learning-based approaches. The dominant approach brought by this progress utilizes learnable features as well as differentiable optimization layers within deep learning frameworks (Chen et al., 2024; Teed et al., 2023). These designs improve convergence and robustness in real-world conditions. However, their performance critically relies on postprocessing techniques such as bundle adjustment or feature matching to refine the estimates of camera poses. The use of such traditional 'feature-engineered' components introduces hand-crafted elements and task-specific hyperparameters to incorporate prior domain knowledge. While proven effective, these practices limit the scaling behavior of models as they do not fully exploit the end-to-end nature of deep learning systems (Tay et al., 2023). Additionally, auxiliary supervision (*e.g.*, by dense optical flow) is often used to support training, further increasing system complexity and limiting training data. Motivated by these limitations, we propose a direct 'end-to-end' pose regression approach that eliminates hand-crafted modules and auxiliary tasks.

Our method draws inspiration from recent successes in end-to-end structured prediction tasks, such as machine translation (Vaswani et al., 2017), image recognition (Dosovitskiy et al., 2021), and object detection (Carion et al., 2020). By learning a mapping directly from video input to camera poses, the model becomes less dependent on predefined priors and can better capture complex temporal and spatial relationships. This design choice aligns with recent evidence (Nguyen et al., 2025) suggesting that reducing inductive bias enables models to generalize more effectively from data. Indeed, recent breakthroughs in image recognition (Dosovitskiy et al., 2021), 2D/3D object detection (Carion et al., 2020; Nguyen et al., 2022), and 3D reconstruction (Wang et al., 2025a; Leroy et al., 2024) increasingly adopt Transformer-based architectures for learning representations with large-scale datasets. We hypothesize that an end-to-end visual odometry framework with a transformer is a key enabler for scalability, leading to improved robustness and accuracy.

In this paper, we introduce the **V**isual **o**dometry **T**ransformer (VoT), a fully end-to-end framework for monocular visual odometry. Our approach formulates visual odometry as a direct pose prediction task, using a transformer-based encoder-decoder architecture. The encoder is initialized with a pre-trained backbone, while the decoder employs both temporal and spatial attention to model interactions across frames. This design implicitly captures motion-specific properties, such as feature matching, without requiring explicit supervision or auxiliary modules.

In our framework, the VoT predicts the sequence of relative camera poses in a single forward pass and is trained end-to-end using loss functions for both translation and rotation. To ensure the validity of rotation predictions, we project them onto the SO(3) manifold and employ a loss function that computes the shortest distance between the predicted and ground-truth rotation matrices. Importantly, our implementation avoids any custom CUDA operations or non-standard layers, improving reproducibility and compatibility with common deep learning frameworks.

We train and evaluate VoT on several large-scale indoor and outdoor datasets where our method achieves competitive accuracy across all metrics. In addition, VoT shows strong scaling behavior and can generalize to unseen datasets. We find that spatial-temporal attention enables the model to implicitly learn global feature interactions, supporting our hypothesis that an end-to-end design trained on extensive data can outperform systems that rely heavily on hand-crafted priors and post-processing. Notably, VoT with the end-to-end design runs more than $3\times$ faster than its counterparts.

## 2 RELATED WORK

**Visual odometry.** Visual odometry systems estimate the position and orientation of a camera using video input. Unlike SLAM which corrects errors via loop closure (Cadena et al., 2016; Campos et al., 2021; Yugay et al., 2024), these systems tend to accumulate camera tracking errors (drift). Many different modalities of visual odometry have been explored by past work, including visual-inertial odometry (Forster et al., 2015; Von Stumberg et al., 2018) and stereo visual odometry (Engel et al., 2014; Wang et al., 2017a). Here, we focus on the monocular case, where the only input is a monocular video stream. Traditional monocular visual odometry methods (Engel et al., 2014; 2018; Campos et al., 2021) are sensitive to illumination and rolling shutter artifacts, and more importantly, cannot adequately estimate the scale of the scene. To overcome these limitations, VoT leverages a large pre-trained encoder and extensive training data for robust generalization.

**Deep monocular visual odometry.** Deep learning has advanced monocular visual odometry in both supervised (Wang et al., 2017b; 2020; Teed & Deng, 2020a; Teed et al., 2023) and unsupervised (Yin & Shi, 2018; Ranjan et al., 2019; Sharma & Ventura, 2019; Li et al., 2020) settings. DeepVO (Wang et al., 2017b) uses recurrent networks for temporal modeling, while SfMLearner (Sharma & Ventura, 2019) jointly learns depth and motion without labels. DPVO (Teed et al., 2023), inspired by RAFT (Teed & Deng, 2020b), combines flow, confidence, and geometry with iterative GRU updates, while LeapVO (Chen et al., 2024) increases robustness in dynamic scenes via keypoints.

Prior work introduced TSFormer (Françani & Maximo, 2025), which predicts camera poses directly from video sequences using an end-to-end approach. However, its performance remains below that of hand-crafted methods (Teed et al., 2023; Chen et al., 2024) due to architecture, small training dataset, and rotation representation. Existing methods generalize poorly because they rely on small, hard-to-scale architectures, complex hand-crafted components, and often require camera parameters,

limiting real-world use. By comparison, VoT uses a scalable end-to-end design that works without camera parameters, enabling deployment across diverse scenarios.

**Large 3D models.** A trend in 3D vision is the development of large-scale models that jointly estimate camera poses and dense geometry (Wang & Agapito, 2025; Wang et al., 2025a;b), or learn versatile representations for downstream tasks (Wang et al., 2024; Leroy et al., 2024). These models are increasingly viewed as foundational, yet they suffer from drift on long video sequences and produce scale-ambiguous poses requiring calibration. In contrast, VoT focuses solely on camera pose estimation, showing improved accuracy and less drift. Moreover, our model is significantly faster.

**Video transformers.** Since their introduction in machine translation (Vaswani et al., 2017), transformers have become the dominant architecture in NLP (Devlin et al., 2019; OpenAI, 2023) and vision (He et al., 2022; Oquab et al., 2024). Central to this framework, self-attention aggregates information across entire sequences, but its quadratic complexity makes direct application to videos inefficient. To mitigate this, prior work has proposed factorizing 3D attention into spatial and temporal components (Zhang et al., 2021) or employing efficient variants such as divided space-time, sparse local-global, and axial attention (Bertasius et al., 2021; Arnab et al., 2021; Zhao et al., 2022). In this work, we adopt the principle of efficient video learning by decoupling self-attention into separate temporal and spatial attention components.

## 3 METHOD

In this section, we describe the architecture of VoT; see Fig. 1 for its overview. It consists of three main components: a pre-trained encoder to extract a compact representation of each frame, a transformer-based decoder, and a small feed-forward network that makes the final pose predictions.

Unlike previous methods such as DVPO (Teed et al., 2023) or LeapVO (Chen et al., 2024), VoT can be implemented in any deep learning framework, thanks to its streamlined architecture. Following the spirit of end-to-end frameworks (Carion et al., 2020), our inference code can be written purely in PyTorch, without additional complexities. We believe that the simplicity and accessibility of our approach will encourage further participation from the research community.

### 3.1 ARCHITECTURE

**Pre-trained encoder.** We utilize pre-trained networks commonly employed in previous studies (Wang et al., 2024; 2025b) for encoding and extracting features from video frames. As the Vision Transformer (ViT) (Dosovitskiy et al., 2021) has become the dominant architecture in computer vision, our pre-trained encoders are based on ViT. Specifically, we denote the sequence of input video frames as $V \in \mathbb{R}^{T \times H \times W \times 3}$, where $T$ is the number of frames in the video and $H, W$ denote the resolution of each frame. Each input image is then tokenized into $(h \cdot w)$ non-overlapping patches with $h = \frac{H}{p}, w = \frac{W}{p}$ and $p$ as patch size. These tokens are processed through a series of ViT layers, producing image features $F \in \mathbb{R}^{T \times (h \cdot w) \times d}$, where $d$ is the hidden dimension of the ViT. The

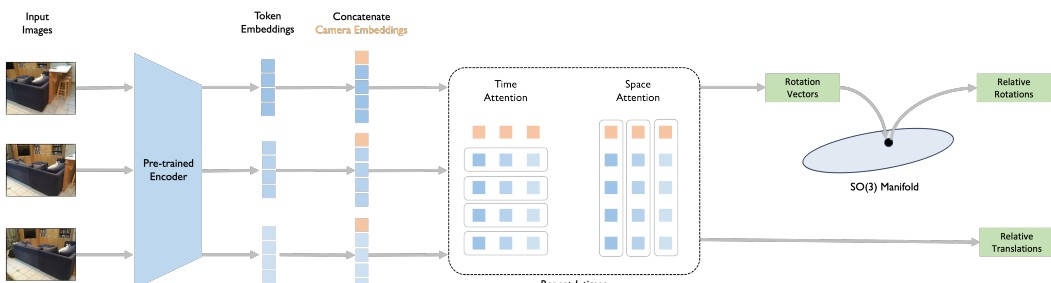

Figure 1: **VoT architecture.** Given multiple input frames, a frozen image encoder extracts per-image token embeddings. Camera embeddings are then concatenated to aggregate the information for camera pose estimation. The embeddings are decoded by $L$ repeating decoder blocks with temporal and spatial attention modules. The rotations are projected onto the SO(3) manifold to ensure valid relative rotations.

ViT layer follows the standard architecture, consisting of a multi-head self-attention module and a feed-forward network. Since the transformer architecture is permutation-invariant, we augment it with sinusoidal position encodings (Vaswani et al., 2017).

**Time-space decoder.** The decoder consists of a stack of $L$ identical layers, each containing three sub-layers. The first sub-layer is a multi-head temporal attention module, followed by a multi-head spatial attention module, and the final sub-layer is a feed-forward network. Unlike standard transformer layers that only use self-attention, the interleaving of temporal and spatial attention within the decoder layers enables each feature in a single frame to efficiently capture long-range dependencies with other features. This ability to learn long-context information effectively is crucial when processing a large number of frames. As we demonstrate in our experiments, using a larger number of frames is key to achieving accurate camera pose predictions. To summarize the information in each frame, we use camera embeddings, which are learnable embeddings in spatial attention sub-layers.

We start by concatenating a camera embedding to each of the image features, resulting in input features to the decoder $F_0 = [ce, F] \in \mathbb{R}^{T \times (h \cdot w + 1) \times d}$ where $ce \in \mathbb{R}^{T \times 1 \times d}$ indicate the camera embeddings. We denote the inputs to the $(n+1)^{\text{th}}$ decoder layer by $F_n \in \mathbb{R}^{T \times (h \cdot w + 1) \times d}$. The $(n+1)^{\text{th}}$ decoder layer then outputs $F_{n+1} \in \mathbb{R}^{T \times (h \cdot w + 1) \times d}$ of the same size. Specifically, the temporal attention performs the scaled dot-product attention in the $i$-th head along the temporal dimension as[1]:

$$\hat{Q} = \hat{K} = \hat{V} = F[:, 1:, :]^\top; \qquad \in \mathbb{R}^{(h \cdot w) \times T \times d} \qquad (1)$$

$$\hat{\text{head}}_i = \text{Attention}(\hat{Q} W_i^{\hat{Q}}, \hat{K} W_i^{\hat{K}}, \hat{V} W_i^{\hat{V}}), \quad \in \mathbb{R}^{(h \cdot w) \times T \times d_h} \qquad (2)$$

where $W_i^{\hat{Q}}, W_i^{\hat{K}}, W_i^{\hat{V}} \in \mathbb{R}^{d \times d_h}$ are the learned projection matrices for query, key, and value. Here, we omit the layer index by treating $F_n = F$.

Note that we omit the camera embedding during the temporal attention computation since attention computation over learnable camera embeddings is redundant. The multi-head temporal attention aggregates $\hat{\text{head}}_{\{1...h\}}$ together and then concatenates the camera embedding.

$$\hat{F} = \text{Concat}(\hat{\text{head}}_i, ..., \hat{\text{head}}_h)^\top W^{\hat{O}}, \qquad \in \mathbb{R}^{T \times (h \cdot w) \times d} \qquad (3)$$

$$\text{TemporalAttention}(\hat{Q}, \hat{K}, \hat{V}) = [F[:,:1,:], \hat{F}], \qquad \in \mathbb{R}^{T \times (h \cdot w + 1) \times d} \qquad (4)$$

where $W_i^{\hat{O}} \in \mathbb{R}^{d_h \times d}$ is the output projection and $F[:,:1,:] \in \mathbb{R}^{T \times 1 \times d}$ indicates the camera embeddings copied from $F$.

Following the temporal attention, we apply the spatial attention along the spatial dimension, including the camera embedding as:

$$\bar{Q} = \bar{K} = \bar{V} = \hat{F} \qquad \in \mathbb{R}^{T \times (h \cdot w + 1) \times d_h} \qquad (5)$$

$$\bar{\text{head}}_i = \text{Attention}(\bar{Q} W_i^{\bar{Q}}, \bar{K} W_i^{\bar{K}}, \bar{V} W_i^{\bar{V}}) \quad \in \mathbb{R}^{T \times (h \cdot w + 1) \times d_h} \qquad (6)$$

where $W_i^{\bar{Q}}, W_i^{\bar{K}}, W_i^{\bar{V}} \in \mathbb{R}^{d \times d_h}$ are the learned projection matrices for query, key, and value.

Similarly, the multi-head spatial attention aggregates $\bar{\text{head}}_{\{1...h\}}$ together.

$$\text{SpatialAttention}(\bar{Q}, \bar{K}, \bar{V}) = \text{Concat}(\bar{\text{head}}_i, ..., \bar{\text{head}}_h) W^{\bar{O}} \qquad \in \mathbb{R}^{T \times (h \cdot w + 1) \times d} \qquad (7)$$

where $W_i^{\bar{O}} \in \mathbb{R}^{d_h \times d}$ is the output projection.

At the end, the camera embedding, $e \in \mathbb{R}^{T \times d}$, will be used to predict the camera poses for each frame.

## 3.2 CAMERA POSES PREDICTION

To regress the relative camera poses, the regression head takes a camera embedding from each frame as its input. A single linear layer then projects the embeddings to the rotation $\mathbb{F}_R \in \mathbb{R}^{(T-1) \times 9}$ and

---

[1]The notation [:, 1:, : ] indicates the submatrix that removes the first column, as in Numpy.

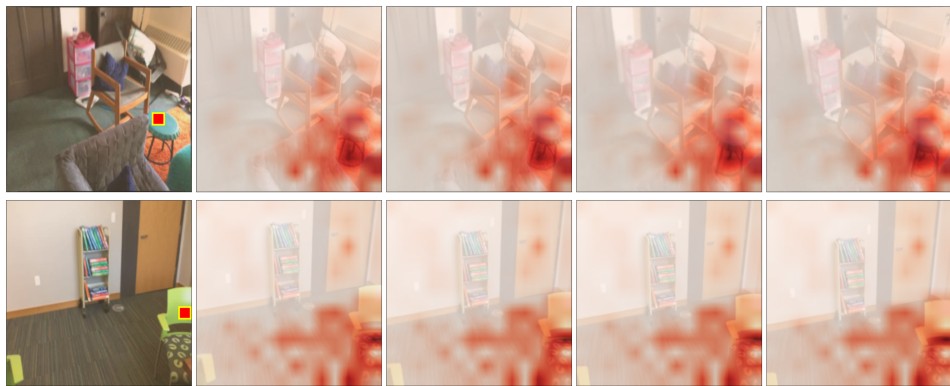

Figure 2: **Attention maps from the VoT decoder.** Each row shows an original image with a selected query (red square), followed by attention maps from the four subsequent frames. To estimate relative camera pose, VoT attends to the related image regions, resembling the behavior of classical keypoint-based odometry methods.

translation $\mathbb{F}_t \in \mathbb{R}^{(T-1)\times 3}$ vectors. Rotation vectors are projected to the closest valid rotation matrix by solving the special orthogonal Procrustes (Brégier, 2021) problem:

$$\text{Procrustes}(\mathbb{F}_R) = \arg\min_{\hat{\mathbb{R}} \in SO(3)} \|\hat{\mathbb{R}} - \mathbb{F}_R\|_F^2.$$

The optimization problem is solved via singular value decomposition of $\mathbb{F}_R$ following Umeyama (1991). $\mathbb{F}_t \in \mathbb{R}^{(T-1)\times 3}$ is used as a translation prediction directly.

**Decoding losses.** The rotation loss, denoted as $\mathcal{L}_{\text{rotation}}(R, \hat{R})$, is the geodesic loss between the predicted rotation $\hat{R}$ and the ground-truth rotation $R$, defined as

$$\mathcal{L}_{\text{rotation}}(R, \hat{R}) = \cos^{-1}\left(\frac{\text{Tr}(R^\top \hat{R}) - 1}{2}\right), \tag{8}$$

and the translation loss, denoted as $\mathcal{L}_{\text{translation}}$, is the L1 loss between the predicted translation $\hat{t}$ and the ground-truth translation $t$:

$$\mathcal{L}_{\text{translation}}(t, \hat{t}) = \|t - \hat{t}\|_1. \tag{9}$$

The final loss for VoT training is a weighted loss between the rotation and translation losses:

$$\mathcal{L} = \lambda \cdot \mathcal{L}_{\text{rotation}} + \gamma \cdot \mathcal{L}_{\text{translation}}. \tag{10}$$

## 4 EXPERIMENTS

**Datasets.** Our primary training leverages ARKitScenes (Baruch et al., 2021), a large-scale dataset, providing rich and diverse indoor scenes. We further incorporate ScanNet (Dai et al., 2017) to broaden indoor coverage and KITTI (Geiger et al., 2012) to expose the model to outdoor pose distributions. Out-of-distribution performance is evaluated on TUM_RGBD (Sturm et al., 2012) which is not included in the training set.

*ARKitScenes* (Baruch et al., 2021) is a large-scale indoor dataset collected with LiDAR-equipped mobile devices, comprising over 5,000 sequences from 1,661 environments and totaling more than 11 million frames, making it one of the most comprehensive resources for indoor 3D scene understanding. To complement it, *ScanNet* (Dai et al., 2017) provides over 2 million frames across 1,613 scenes. For outdoor settings, *KITTI* (Geiger et al., 2012) offers RGB images and ground-truth poses collected with GPS and LIDAR. Finally, *TUM_RGBD* (Sturm et al., 2012) features indoor recordings with motion-capture ground truth. The datasets exhibit variation in camera calibration, poses, image resolution, as well as real-world artifacts such as rolling shutter, motion blur, and lighting variation.

**A note on alignment**. While most existing methods report evaluation metrics after applying rigid alignment and scale correction to the predicted trajectory, we believe this practice can be misleading

| Method | w/o Bundle Adjustment | w/o Camera Parameters | ATE [m] ↓ | ARE [deg] ↓ | RTE [m] ↓ | RRE [deg] ↓ |
|---|---|---|---|---|---|---|
| ORB-SLAM3 (Campos et al., 2021) | ✗ | ✗ | 2.58 | 111.58 | 0.17 | 4.01 |
| DPVO (Teed et al., 2023) | ✗ | ✗ | 5.48 | 18.68 | 0.04 | 0.48 |
| LeapVO (Chen et al., 2024) | ✗ | ✗ | 28.31 | 49.25 | 0.62 | 1.09 |
| TSFormer (Françani & Maximo, 2025) | ✓ | ✓ | 285.22 | 97.53 | 0.52 | 0.62 |
| CUT3R (Wang & Agapito, 2025) | ✓ | ✓ | 2.42 | 97.70 | 0.12 | 4.86 |
| VGGT (Wang et al., 2025a) | ✓ | ✓ | 2.94 | 92.37 | 0.15 | 4.81 |
| Mast3r-SLAM-VO (Murai et al., 2025) | ✓ | ✓ | 0.60 | 14.40 | 0.36 | 17.90 |
| VoT (Ours) | ✓ | ✓ | **0.41** | **10.10** | **0.04** | **0.25** |

Methods are evaluated using all frames of the test split without alignment to the ground-truth.

Table 1: **Pose estimation metrics on ARKitScenes *indoor* settings.** VoT shows superior performance across all metrics without relying on post-optimization and camera parameters.

for real-world applications, where ground-truth trajectories are unavailable for alignment. Although early approaches relied on this convention due to the inability to robustly estimate the scale of a scene, the assumption is less justified in the context of modern large-scale 3D models. Therefore, all experimental results are based on *unaligned* predictions.

We evaluate visual odometry using Absolute Translation Error (ATE) and Absolute Rotation Error (ARE) (Teed et al., 2023; Chen et al., 2024). ATE is the RMSE of translation between estimated and ground-truth trajectories, while ARE is the RMSE of angular orientation differences. In addition, we report Relative Translation Error (RTE) and Relative Rotation Error (RRE), which measure the drift in translation and rotation over unit trajectory segments. In the tables, results are color-coded to indicate ranking: **best**, second-best, and third-best.

**Baselines**. We compare our method against state-of-the-art visual odometry models (Teed et al., 2023; Chen et al., 2024). For DPVO (Teed et al., 2023), results are averaged over three runs with different random seeds to account for variability. We further benchmark against recent large-scale 3D models that directly predict camera poses (Wang et al., 2025a; Wang & Agapito, 2025). Because these models cannot process long video sequences in a single pass, inputs are split into temporally continuous chunks, with maximum lengths of 30 and 90 frames on our GPUs, respectively; the predicted poses are then concatenated to form the full trajectory. We also compare with a classical (Campos et al., 2021) and a state-of-the-art (Murai et al., 2025) monocular SLAM systems, both using only RGB input. For fairness, loop closure is disabled during evaluation. Finally, we include a recent end-to-end visual odometry model (Françani & Maximo, 2025) in our comparisons.

**Implementation details.** We use a frozen CroCo (Weinzaepfel et al., 2022) backbone trained within the DUST3R (Wang et al., 2024) framework, consisting of 300 million parameters. We employ 12 alternating time-space attention blocks, totalling 200 million parameters, for the decoder. Our model is trained with the AdamW (Loshchilov & Hutter, 2019) optimizer for 300 epochs. We adopt a cosine learning rate schedule with an initial learning rate of 0.00001 and a warmup phase of 30 epochs. Our model takes 12 input views, sampled at intervals of 3 frames. Input frames are resized to a resolution of 224×224. Training runs on 12 NVIDIA RTX H100 GPUs for 5 days.

### 4.1 VISUAL ODOMETRY RESULTS

**VoT is competitive without bells and whistles.** As shown in Tabs. 1 to 3, VoT achieves competitive performance on both indoor and outdoor datasets, despite not relying on hand-crafted components. This is particularly notable in real-world settings, where *unaligned trajectory* metrics are more realistic due to the lack of ground-truth poses.

Large 3D models such as CUT3R (Wang & Agapito, 2025) and VGGT (Wang et al., 2025a), although trained on diverse multi-task 3D datasets - including both ScanNet and ARKitScenes - fail to generalize effectively. CUT3R and VGGT exhibit substantial drift in long sequences, while Mast3r-SLAM-VO (Murai et al., 2025) suffers from scale ambiguity and sparse predictions, resulting in degraded ATE and failure on the KITTI dataset.

| Method | w/o Bundle Adjustment | w/o Camera Parameters | ATE [m] ↓ | ARE [deg] ↓ | RTE [m] ↓ | RRE [deg] ↓ |
|---|---|---|---|---|---|---|
| ORB-SLAM3 (Campos et al., 2021) | ✗ | ✗ | 1.91 | 100.42 | 0.04 | 3.75 |
| LeapVO (Chen et al., 2024) | ✗ | ✗ | 10.84 | 43.60 | 0.09 | 0.43 |
| DPVO (Teed et al., 2023) | ✗ | ✗ | 1.75 | 5.91 | 0.02 | 0.34 |
| TSFormer (Françani & Maximo, 2025) | ✓ | ✓ | 285.22 | 97.53 | 0.52 | 0.62 |
| CUT3R (Wang & Agapito, 2025) | ✓ | ✓ | 4.85 | 135.52 | 0.02 | 0.65 |
| VGGT (Wang et al., 2025a) | ✓ | ✓ | 1.56 | 31.73 | 0.06 | 1.42 |
| Mast3r-SLAM-VO (Murai et al., 2025) | ✓ | ✓ | 0.99 | 8.40 | 0.58 | 6.96 |
| VoT (Ours) | ✓ | ✓ | 0.32 | 7.67 | 0.01 | 0.34 |

Methods are evaluated using all frames of the test split without alignment to the ground-truth.

Table 2: **Pose estimation metrics on ScanNet *indoor* settings.** VoT shows competitive performance without relying on bundle adjustment or requiring camera parameters. Moreover, our method accurately predicts absolute poses, as demonstrated by the low ATE.

| Method | w/o Bundle Adjustment | w/o Camera Parameters | ATE [m] ↓ | ARE [deg] ↓ | RTE [m] ↓ | RRE [deg] ↓ |
|---|---|---|---|---|---|---|
| ORB-SLAM3 (Campos et al., 2021) | ✗ | ✗ | 217.06 | 51.23 | 1.05 | 1.20 |
| DPVO (Teed et al., 2023) | ✗ | ✗ | 194.55 | 0.94 | 0.85 | 0.06 |
| LeapVO (Chen et al., 2024) | ✗ | ✗ | 211.80 | 39.78 | 0.94 | 0.96 |
| TSFormer (Françani & Maximo, 2025) | ✓ | ✓ | 82.05 | 22.84 | 0.23 | 0.24 |
| CUT3R (Wang & Agapito, 2025) | ✓ | ✓ | 112.36 | 22.07 | 0.71 | 0.58 |
| VGGT (Wang et al., 2025a) | ✓ | ✓ | 205.67 | 16.32 | 2.13 | 0.25 |
| Mast3r-SLAM-VO (Murai et al., 2025) | ✓ | ✓ | – | – | – | – |
| VoT (Ours) | ✓ | ✓ | 58.03 | 11.69 | 0.61 | 0.22 |

Methods are evaluated using all frames of the test split without alignment to the ground-truth.

Table 3: **Pose estimation metrics on KITTI *outdoor* settings.** VoT is capable of modeling very different outdoor camera pose distribution, showing competitive performance compared to counterparts. "–" indicates the failure of a method.

Among odometry-specific methods, VoT exhibits strong generalization across datasets. In contrast, LeapVO (Chen et al., 2024), trained solely on synthetic data, struggles on real-world videos. DPVO (Teed et al., 2023) achieves competitive results on ScanNet but deteriorates on ARKitScenes and KITTI, particularly in scale estimation (ATE), limiting its applicability in the wild. ORB-SLAM3 (Campos et al., 2021) fails to reliably estimate scale, leading to poor performance even with loop closure and bundle adjustment enabled. TSFormer (Françani & Maximo, 2025), trained on a small dataset (*i.e.*, KITTI (Dosovitskiy et al., 2021)), without strong encoders or robust rotation representations, fails to predict accurate trajectories.

| Method | w/o Bundle Adjustment | w/o Camera Parameters | ATE [m] ↓ | ARE [deg] ↓ | RTE [m] ↓ | RRE [deg] ↓ |
|---|---|---|---|---|---|---|
| ORB-SLAM3 (Campos et al., 2021) | ✗ | ✗ | 1.65 | 76.14 | 3.16 | 0.03 |
| DPVO (Teed et al., 2023) | ✗ | ✗ | 0.94 | 3.02 | 0.07 | 0.44 |
| LeapVO (Chen et al., 2024) | ✗ | ✗ | 1.06 | 7.36 | 0.03 | 0.56 |
| TSFormer (Françani & Maximo, 2025) | ✓ | ✓ | 135.49 | 84.29 | 0.60 | 2.78 |
| CUT3R (Wang & Agapito, 2025) | ✓ | ✓ | 0.87 | 29.40 | 0.14 | 1.41 |
| VGGT (Wang et al., 2025a) | ✓ | ✓ | 1.77 | 31.66 | 0.08 | 1.82 |
| Mast3r-SLAM-VO (Murai et al., 2025) | ✓ | ✓ | 2.31 | 145.22 | 0.08 | 2.02 |
| VoT (Ours) | ✓ | ✓ | 0.74 | 28.95 | 0.02 | 2.55 |

Methods are evaluated using all frames of the test split without alignment to the ground-truth.

Table 4: **Pose estimation metrics on TUM_RGBD *out-of-distribution* setting.** While optimization-based methods shows strong performance in terms of ARE, VoT is capable of generalizing to unseen data and camera parameters and shows superior performance in ATE and RTE.

## 4.2 ABLATION STUDIES

We conduct ablation studies to assess key design choices and analyze scaling trends with respect to model capacity and dataset size. For architectural ablations (e.g., attention mechanisms, backbone, and rotation representation), we use two input views with a stride of 10 and train for 150 epochs. For scaling experiments (e.g., varying dataset size), models are trained to convergence for 300 epochs to accurately reflect scalability. All ablations are evaluated on the ScanNet (Dai et al., 2017) dataset.

**VoT generalizes to unseen datasets and camera parameters.** Tab. 4 illustrates how VoT performs on the dataset not seen during training. Moreover, the camera parameters on the dataset are drastically different from the ones in the training set.

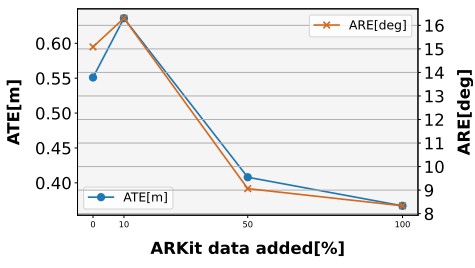
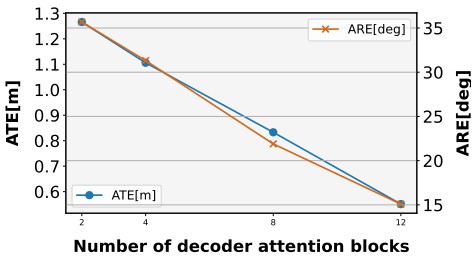

(a) Scaling behavior w.r.t. training data        (b) Scaling behavior w.r.t. model capacity

Figure 3: **Scaling behavior of VoT.** As the model scales in (a) training data (proportion of ARKitScenes data added to ScanNet) and (b) model capacity (number of decoder layers), absolute translation and rotation errors decrease. This suggests that VoT exhibits robust scaling behavior.

**VoT exhibits robust scaling behavior.** Fig. 3 illustrates how VoT scales with respect to training data size and model capacity. VoT shows consistent performance improvements with both increased training data and a larger number of trainable parameters. Due to computational limits, we cap the number of decoder layers at 12. We analyze the scaling behavior with respect to encoder size and number of input views in the Supplementary Material.

| Method | ATE[m]↓ | ARE[deg]↓ | RTE[m]↓ | RRE[deg]↓ |
|---|---|---|---|---|
| MAE | 1.26 | 28.30 | 0.08 | 2.22 |
| DinoV2 | 1.05 | 31.52 | 0.07 | 2.09 |
| CroCoV2 | 1.02 | 32.58 | 0.07 | 2.46 |
| CroCoV2* | **0.55** | **15.09** | **0.04** | **1.25** |

Table 5: **Backbone ablation.** * denotes the model trained within the Dust3r (Wang et al., 2024) framework with 3D supervision. The results highlight the critical role of training data in determining feature extractor performance, with geometry-focused 3D supervision yielding better results across all metrics.

**Pre-training data is the key determinant of backbone performance.** We compare features extracted from various pre-trained backbones in Tab. 5. While DINOv2 (Oquab et al., 2024) and CroCoV2 (Weinzaepfel et al., 2022) yield comparable results, the CroCoV2 encoder trained within the Dust3R (Wang et al., 2024) framework that uses geometry-focused 3D data - achieves substantially better performance (i.e., approximately 2× lower error across all metrics). These results suggest that the quality and relevance of pre-training data, rather than backbone architecture, are critical for learning effective features in visual odometry.

| Attention Type | ATE[m]↓ | ARE[deg]↓ | RTE[m]↓ | RRE[deg]↓ | GFLOPs↓ |
|---|---|---|---|---|---|
| Full Attention | 1.57 | 60.24 | 0.09 | 4.04 | 380 |
| Time-Space Attention | 1.27 | 35.68 | 0.08 | 2.22 | 163 |

Table 6: **Ablation of attention mechanisms.** Time-space attention outperforms full attention while being considerably more efficient. We attribute this to its ability to capture temporal changes by linking spatially corresponding patches across frames.

| Rotation Rep. | ATE[m] ↓ | ARE[deg] ↓ | RTE[m] ↓ | RRE[deg] ↓ |
|---|---|---|---|---|
| Euler Angles | 0.78 | 20.64 | 0.05 | 1.47 |
| Quaternion | 0.68 | 18.92 | 0.05 | 1.44 |
| Plucker Rays (Zhang et al., 2024) | 0.66 | 16.11 | 0.05 | 1.48 |
| **Rotation Matrix** | **0.55** | **15.09** | **0.04** | **1.25** |

Table 7: **Ablation of rotation representations.** Projecting predictions onto the nearest valid rotation matrix under the Frobenius norm yields the best performance.

**Time-space attention achieves superior performance with lower computational cost.** Tab. 6 compares full self-attention and time-space attention. Time-space attention not only reduces computational cost but also consistently improves accuracy. We attribute this to its matching-like spatial mechanism (see Fig. 2), where patches at corresponding locations across frames exchange temporal information, enhancing motion-aware feature encoding.

**SO(3) projection enhances rotation accuracy.** In Tab. 7, we evaluate different rotation representations. Projecting outputs onto the nearest valid rotation matrix on the SO(3) manifold using the Frobenius norm consistently yields the best results, highlighting its suitability for our formulation.

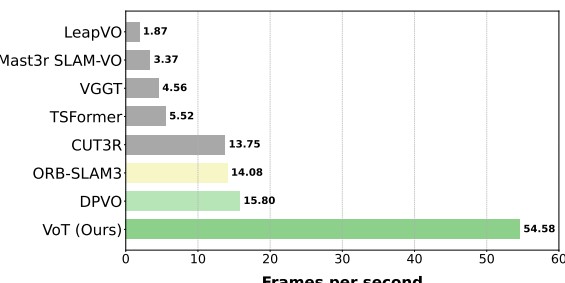

Figure 4: **Runtime analysis.** VoT demonstrates a considerable speedup compared to other methods, including large 3D models like VGGT (Wang et al., 2025a) and odometry methods such as DPVO (Teed et al., 2023). Different from end-to-end TS-Former (Françani & Maximo, 2025), VoT does not do overlapping relative pose averaging making inference significantly faster. All methods are profiled using the same machine with an NVIDIA L4.

**VoT delivers real-time prediction.** To compare the speed between approaches, we measure their runtime on ScanNet (Dai et al., 2017) *scene0000_00* using the same environment. As shown in Fig. 4, VoT achieves a considerably faster runtime, approximately 3× speedup, compared to existing methods. This efficiency stems from its compact, end-to-end architecture, in contrast to larger 3D models. Moreover, unlike LeapVO, DPVO, and Mast3R-SLAM-VO, VoT does not rely on additional processing steps or bundle adjustment, contributing to both its speed and strong performance.

**Limitations and future work.** We do not claim VoT can generalize to all videos. Since it is trained on static environments, performance may be limited in dynamic settings. Future improvements could come from scaling to more diverse datasets collected with different devices and careful calibration, as well as expanding coverage to a wider variety of scenes. Employing systematic data curation and incorporating larger models or advanced pre-trained components are also promising directions that could enhance both generalization and overall performance.

## 5 CONCLUSION

We presented VoT, an end-to-end method for visual odometry systems based on transformers for direct relative pose prediction. The approach achieves strong results compared to modern state-of-the-art odometry pipelines and large 3D models on the challenging indoor and outdoor datasets. VoT has a flexible architecture easily extensible to various backbones, with competitive results. Importantly, it can generalize to out-of-distribution camera parameters and image data. It exhibits strong scaling behavior, indicating that its performance could be further enhanced with sufficient data and computational resources. Finally, it is considerably faster than the baselines, being more compact than large 3D models, and, unlike other odometry systems, it does not require test-time optimization. We anticipate that future work will further advance end-to-end odometry systems, extending their scalability to a broader range of domains.

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

# A    APPENDIX

We present further ablation studies and visual examples. Specifically, we test how well VoT scales with different backbone sizes and numbers of input views. We also compare the trajectories predicted by VoT to those from baseline methods, demonstrating the effectiveness of our model.

**Performance of VoT with number of input views.** Figure 5 illustrates how VoT, trained on the full dataset, scales with the number of input views. While the performance plateaus in terms of absolute metrics, it consistently improves in relative metrics. Due to computational constraints, the number of input views in our experiments is limited to a maximum of 12.

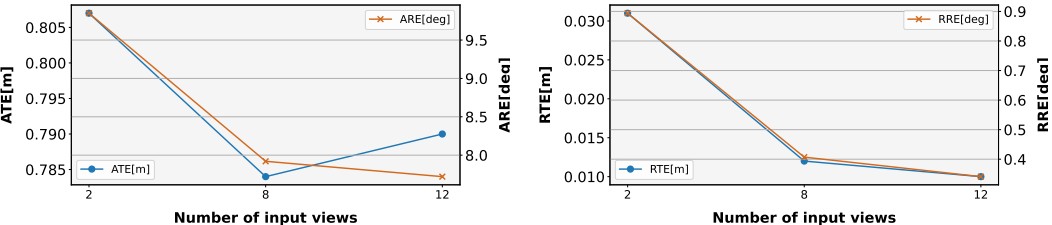

Figure 5: **Performance of VoT with number of input views on ScanNet (Dai et al., 2017) dataset.** As the number of input views increases, absolute translation and rotation errors initially decrease before plateauing. In contrast, relative metrics continue to improve steadily, highlighting the potential of VoT when using a long-range input sequence of views.

**Performance of VoT with backbone size.** In Fig. 6, we show how the performance of VoT varies with the size of the backbone. Since CroCO (Weinzaepfel et al., 2022) is only available in a single size, we perform ablation using Dino-V2 (Oquab et al., 2024) backbones. We observe that performance improves as the size of the backbone is increased. Due to computational limits, we cap the backbone size at DinoV2 (Oquab et al., 2024) large.

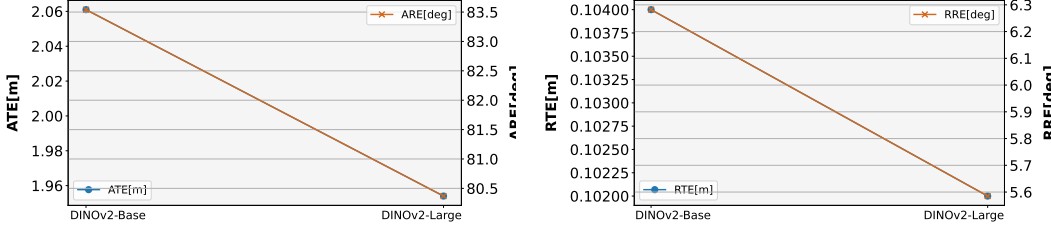

Figure 6: **Performance of VoT with backbone size on ScanNet (Dai et al., 2017) dataset.** As the number of backbone parameters increases, the model's performance improves.

**Datasets.** ScanNet Dai et al. (2017) is an extensive RGB-D corpus designed for 3D scene under-standing in indoor environments, containing over 2 million RGB-D frames from 1,613 unique scenes. Some sequences contain invalid camera poses; in such cases, we use all frames up to the first invalid pose, filtering out 20% of the frames. Additionally, unrealistic jumps in camera trajectories, such as translations spanning several meters between consecutive frames, may occur. To mitigate this, frames with translation magnitudes exceeding 1.5 meters are excluded during training. Our method is evaluated on the entire test split of ScanNet, comprising 98 complete video sequences, using only RGB input. We train our method on KITTI Geiger et al. (2012) by putting 70% of the publicly available data in the training set, and the rest as a hold-out set for testing.

**Qualitative Results.** In Fig. 7, we compare several visual odometry methods on representative scenes from the ScanNet (Dai et al., 2017) test set. Each method is evaluated on the entire video sequence using its default settings. All trajectories are visualized without alignment to the ground-truth poses

to reflect real-world deployment conditions better. VoT consistently demonstrates accurate and robust camera pose estimation across all evaluated scenes.

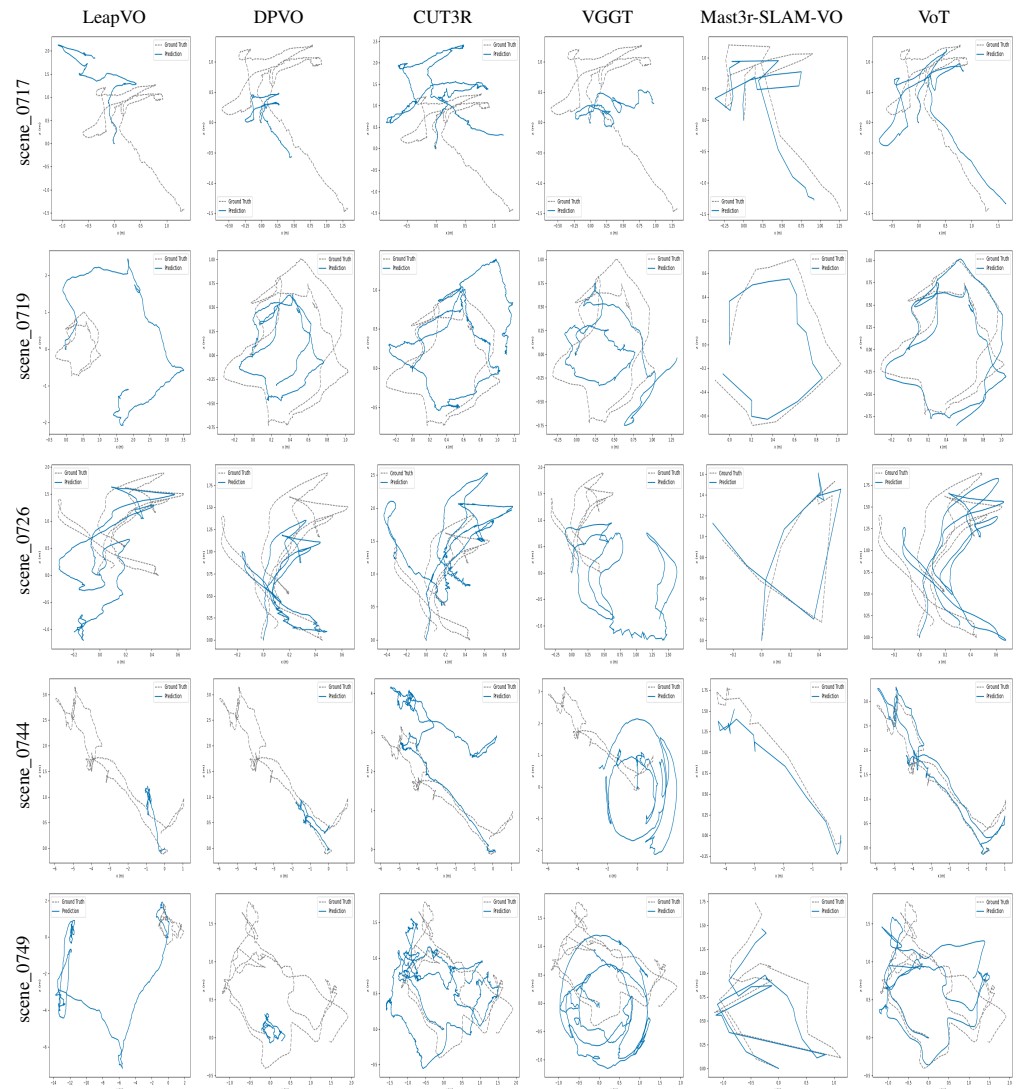

Figure 7: **Qualitative comparisons.** Each column shows a method, each row a different scene from the ScanNet (Dai et al., 2017) test set. We evaluate all methods on the *whole* trajectory *without* aligning predictions to ground truth, reflecting realistic deployment conditions. LeapVO and DPVO fail to recover scale, while CUT3R and VGGT exhibit significant drift. Mast3r-SLAM-VO estimates poses only at sparse keyframes. VoT consistently achieves robust and accurate reconstructions across diverse scenarios.

