# OpenReview forum: "Visual Odometry with Transformers"
_ICLR.cc/2026/Conference — ICLR 2026 Conference Withdrawn Submission_

### Official Review · Reviewer_S7Na · 2025-10-27

**Soundness:** 3
**Presentation:** 4
**Contribution:** 2
**Rating:** 2
**Confidence:** 2

**Summary:**

This work proposes a ViT-based visual odometry method. The key idea is to add a learnable camera embedding into a pre-trained ViT architecture, and train the model to directly regress camera poses end-to-end, without depth or other 3D info for supervision.

**Strengths:**

The framework is simple, including a simple ViT backbone and simple pose-only supervision.

Paper writing is clear and eazy to understand.

Comprehensive ablations are conducted.

**Weaknesses:**

From the methodology perspective, the proposed method is essentially not too different from VGGT/MapAnything type of architectures with multi-tasks removed from supervision and focus solely on regressing the camera poses. Though efficient spatial-temporal attention is applied, these are common technologies in efficient video understanding/generation frameworks. Hence, I feel like novelty is an issue, and it is unsurprising to me that such architecture would work for pose estimation given that it already works for both pose and 3D estimation.

In the experiment, the comparisons are a little bit unfair. For example, the advantage of transformers-based frameworks are generalization, while most of the comparison tables are actually in-domain for the proposed method while potentially zero-shot for other methods, such as DPVO. Even so, the proposed method is not much better than DPVO and sometimes performs much worse on zero-shot datasets like TUM-RGBD for the rotation metrics. This indicates that the performance of the proposed method is limited and the exploration of data-centric approaches for generalization is also limited, which cannot convince me that the proposed method is ready to replace modularized approaches like DPVO that are potentially much more efficient.

Btw, the ATE is not making sense since for most of the models evaluated, they did not perform metric scale pose estimation.

**Questions:**

NA

---

### Official Review · Reviewer_f6nn · 2025-10-29

**Soundness:** 2
**Presentation:** 2
**Contribution:** 2
**Rating:** 2
**Confidence:** 3

**Summary:**

The work presents an end-to-end transformer-based model for monocular visual odometry (VoT) that directly regresses relative camera poses from RGB sequences without auxiliary geometry estimation or bundle adjustment. The method focuses on simplicity, scalability, and speed. It claims superior generalization and runtime efficiency compared to both classical and modern odometry frameworks across a diverse set of datasets.

**Strengths:**

1. The end-to-end method avoids custom operations, encouraging reproducibility. The transformer design is elegant and conceptually clear.
2. The work shows an impressive runtime performance of 55 FPS, which surpasses other baselines by a large margin.
3. Comparison of different image encoders is insightful and can be helpful for researchers working in the area of ego-motion estimation.
4. The method shows superior performance on sequences from the unseen TUM RGB-D dataset.

**Weaknesses:**

1. The paper explicitly avoids any alignment between predicted and ground-truth trajectories, arguing that "this practice can be misleading for real-world applications" (lines 269 - 288). However, this design choice does not consider that other baselines, such as DPVO, ORB-SLAM3, LeapVO, do not estimate metric scale from monocular input by design. Evaluating all methods without the scale alignment, therefore, unfairly penalizes them. Evaluation protocols in visual odometry typically separate scale-invariant and scale-aware metrics to allow fair comparison between systems with different scale assumptions (Sturm et al., 2012; Geiger et al., 2012; Zhou et al., 2017). Therefore, the work should conduct an evaluation with alignment to ensure fairness towards other baselines that have been proposed before the era of "modern large-scale 3D models".

2. Generalization is assessed only on a single unseen dataset (TUM-RGBD), which has relatively constrained camera motion and scene variation. More extensive testing on, e.g., ETH3D, TartanAir, EuRoC, 7-Scenes, or other OOD datasets would be needed to substantiate the "robust generalization" claim.

**Questions:**

1. The paper repeatedly implies that VoT excels in predicting absolute scales; however, no analysis is provided on how the metric scale emerges during training from non-calibrated monocular input.
2. Attention maps in Figure 2 look rather unconvincing due to the offset in the epicenter that should have aligned well with the matched regions. Further investigation into the quality of emerged matches or comparison against Transformer-based keypoint matching methods (e.g., TransforMatcher, Kim et al. 2022) is desired to confirm their meaningfulness.

---

### Official Review · Reviewer_xr74 · 2025-11-01

**Soundness:** 3
**Presentation:** 3
**Contribution:** 2
**Rating:** 2
**Confidence:** 5

**Summary:**

This paper proposes VoT (Visual Odometry Transformer), an end-to-end framework for monocular visual odometry (VO) that eliminates hand-crafted components such as bundle adjustment and feature matching. VoT uses a pre-trained Vision Transformer (ViT) to extract frame features and a transformer decoder with temporal-spatial attention to capture global frame interactions. Trained only with camera pose supervision, experimental results show VoT outperforms traditional VO methods (ORB-SLAM3) and large 3D models (e.g., CUT3R, VGGT) across metrics while running ≥3× faster.

**Strengths:**

1. End-to-End Design: VoT performs pose estimation in an end-to-end manner, eliminating hand-crafted modules and auxiliary supervision. This results in an elegant, simple structure with strong scalability, reducing reliance on calibration, hyperparameter tuning, and post-processing.
2. Rigorous Experiments: VoT is thoroughly evaluated on indoor (ARKitScenes, ScanNet), outdoor (KITTI), and out-of-distribution (TUM_RGBD) datasets using unaligned trajectories for comparison.

**Weaknesses:**

1.  The experimental comparisons are fundamentally flawed:
All results are reported on unaligned predictions, which is unreasonable. Monocular, BA-based methods like DPVO and ORB-SLAM3 have no inherent scale, so evaluating their trajectories without alignment is meaningless. In our tests, after scale alignment on KITTI, DPVO and ORB-SLAM3 produce highly accurate trajectories. To provide a meaningful comparison, aligned results must be reported; otherwise, the apparent advantage of end-to-end methods is merely due to fitting the scene scale, not superior pose estimation.
2. Poor Performance:
Even when benefiting from the end-to-end fitting of scene scale, VoT still shows clear weaknesses in rotation accuracy. For instance, on KITTI, its rotational precision is far below that of DPVO, and this gap would likely widen after alignment, both in terms of ATE and ARE. Outdoor scenarios remain inherently challenging for learning-based pose estimation, and this paper does not demonstrate any improvement in this regard.
3. Lack of novelty:
The idea of end-to-end pose estimation is no longer novel, and the paper fails to convincingly demonstrate its superiority in terms of performance.

**Questions:**

1. Experimental comparisons: aligned results must be reported.
2. Poor performance on rotation accuracy on KITTI, outdoor scenarios remain inherently challenging for learning-based pose estimation, and this paper does not demonstrate any improvement in this regard.

---

### Official Review · Reviewer_ZqR3 · 2025-11-02

**Soundness:** 1
**Presentation:** 2
**Contribution:** 1
**Rating:** 2
**Confidence:** 5

**Summary:**

In this paper, the authors propose a new method built completely on visual transformer for end-to-end visual odometry (VO).

More specifically, the proposed method adopts separate temporal and spatial attentions to handle temporal and spatial correlations among input sequences, separately. Compared with global attention, this design is more efficient.

Moreover, the proposed method predicts the rotation on SO3 as opposed Euler angles or Quaternion to improve the pose accuracy.

Experiments on public ARKitSense, KITTI, and ScanNet datasets demonstrate that the proposed gives lower position and rotation errors on unaligned predictions than prior methods such as VGGT.

Ablation studies showcase the influence different number of attention blocks, feature encoders, and rotation representations on final pose estimation.

**Strengths:**

1.	The proposed method uses very similar architecture with VGGT, so it is easy to follow and implement.

2.	The paper is easy to read.

3.	Experiments on three public datasets with metrics of ATE, ARE, RTE, RRE are good.

4.	Ablation studies contain experiment’s on the number of attention blocks, different rotation representations, and different feature encoders are useful.

**Weaknesses:**

The paper has several weaknesses in terms of both method and evaluation.

1.	Novelty. The key novelty of this paper is the temporal and spatial attention which are very close to the design of VGGT, so  I do not think this paper has clear novelties.

2.	Evaluation on unaligned trajectories does not make any sense.

The proposed method is probably supervised with metric poses in the training process, which enables it to predict metric-aware translations. However, many prior methods like VGGT, CUT3R are supervised with normalized poses, which are scale-invariant. Therefore, evaluations on unaligned predictions shows that the proposed method has much small translation errors than prior methods. If we only look at the rotation metrics (RRE – relative rotation error), the proposed method does not have obvious improvements. This can also be observed from the trajectories shown in Figure 7.

In conclusion, the claim that the proposed method outperforms is not supported.

I would like to the evaluation on aligned trajectories.


3.	Effectiveness of temporal and spatial attention.

One key contribution of this paper is the separate temporal and spatial attentions. The spatial attentions is the same as the frame attention in VGGT. The temporal attention performs attention on the same patches (at location [u, v]) across all input frames.

First of all, although the temporal attention reduces the processing time, it relies on a strong assumption that the patch at the same location [u, v] share the information, which is not true in real cases as camera are moving. In theory, this design has flaws.

Secondly, it is confusing that Table 6 in the ablation study experiment shows the temporal and spatial attention works better than full attention as the full attention is more straightforward way of propagating information across temporal dimension and building correspondences across frames. Besides, did the author do both frame attention and full attention for setting of “Full Attention” or just a full attention?

4.	Efficacy of SO3 rotation representation. Table 7 in the ablation study shows the Rotation Matrix gives better performance than Quaternion and other representations. However, Table 3 and 4 demonstrate that VGGT which uses Quaternion gives close even better rotation performance than the proposed method on metric RRE (this metric is not influenced by the alignment).

**Questions:**

1.	Table 5 shows that CroCoV2* pretrained with 3D data in DUST3R gives the best poses. Can we take the proposed method as a method only supervised with poses?

2.	did the author do both frame attention and full attention for setting of “Full Attention” or just a full attention?

3.	In the pose evaluation process, how many frames are evaluated each time for the proposed and previous methods? Figure 7 shows the full trajectories, how did the author stich image clips into a full sequence?


4.	For the task, numbers usually do not give us the understanding of how well an approach works. Instead, the trajectories do. I would suggest moving Figure 7 into the main paper as opposed to the appendix.

---

### Note · Authors · 2025-11-12

I have read and agree with the venue's withdrawal policy on behalf of myself and my co-authors.